# Gesture Area Coverage to Assess Gesture Expressiveness and Human-Likeness

Rodolfo L. Tonoli
Paula D. P. Costa
r105652@dac.unicamp.br
paulad@unicamp.com
Department of Computer Engineering and Automation,
FEEC - Unicamp
Campinas, SP, Brazil

Leonardo B. de M. M. Marques*
Lucas H. Ueda*
lmenezes@cpqd.com.br
lhueda@cpqd.com.br
CQPD
Campinas, SP, Brazil

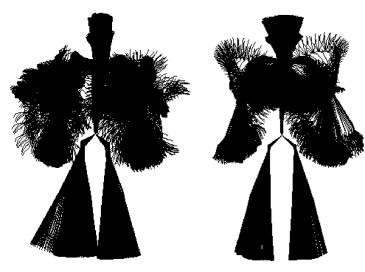 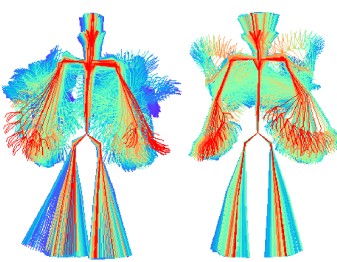 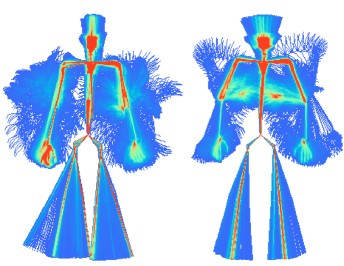

Figure 1: Gesture Area Coverage (GAC) of two motion sequences from the TWH dataset. From left to right: GAC; black represents the total area covered in each sequence. Colored GAC for velocity in each frame; red is faster than blue. Colored GAC for gesture occurrence frequency; red is more frequent than blue.

## Abstract

This work introduces the analyses of Gesture Area Coverage (GAC) for evaluating the expressiveness of co-speech gestures. GAC explicitly considers the spatial coverage of gestures within motion sequences, an aspect existing metrics neglect. We employ a set of metrics based on GAC to compare motion sequences across two key scenarios. First, to differentiate between distinct gesture styles and a neutral baseline, and second, to assess conditions from the GENEA Challenge 2023, a benchmark for gesture generation systems. In particular, our findings reveal that one of these metrics, the Dice Score, has a stronger correlation with human-likeness ratings compared to the Fréchet Gesture Distance.

## CCS Concepts

• **Computing methodologies** → **Motion processing**; • **General and reference** → *Metrics*.

## Keywords

Co-speech gestures, Expressive gestures, Objective evaluation

*Also with Department of Computer Engineering and Automation, FEEC - Unicamp.

**ACM Reference Format:**
Rodolfo L. Tonoli, Paula D. P. Costa, Leonardo B. de M. M. Marques, and Lucas H. Ueda. 2024. Gesture Area Coverage to Assess Gesture Expressiveness and Human-Likeness. In *INTERNATIONAL CONFERENCE ON MULTIMODAL INTERACTION (ICMI Companion '24), November 4–8, 2024, San Jose, Costa Rica.* ACM, New York, NY, USA, 5 pages. https://doi.org/10.1145/3686215.3688822

## 1 Introduction

Gestures are an important aspect of human communication. Co-speech gestures are movements of the hands, arms, head, and body when talking. For the past years, data-driven gesture generation has gained attention, intending to create digital humans capable of interacting as naturally as possible with the user. However, human gestures' unstructured and highly variable nature introduces several challenges for those systems.

Gestures carry expressive information similar to verbal and other nonverbal behaviours. Behaviour expressiveness can be defined as the dynamic variation of the behaviour [15]. It has been shown that personality affects how people gesticulate [6], and that gestures can aid personality perception [14]. Thus, envisioning characters with unique and memorable personalities, gesture generation systems should not only model the human gesture distribution but also transform movement parameters to match the intended expressiveness.

Another challenge of gesture generation systems is the lack of metrics to evaluate the motion output during development. Researchers often rely on perceptual evaluations to assess human-likeness and appropriateness of gestures to the input speech. However, user studies are time-consuming and expensive. Although

some objective metrics have been proposed, most fail to accommodate the one-to-many nature of gestures, while others tend to focus on motion dynamics, e.g., mean velocity and acceleration, but fail to recognize the anatomic correctness of output poses.

## 2 Related Works

Perhaps the most straightforward metric to compare a motion sequence to a reference are objective metrics like Mean Squared Error (MSE) or Mean Average Error (MAE). While they provide a basic measure of the difference between gestures, they treat the reference as the absolute truth. This approach disregards the inherent variability of human gestures. The same expressiveness or intent can be achieved with variations in the movements while still being natural, such as performing a gesture with the other arm. MSE and MAE penalize these variations and neglect the dynamics of movement, such as speed, acceleration, and deceleration. Other metrics to evaluate the appropriateness and semantic relationship to input speech have been proposed, such as BeatAlign and SRGR [12]. However, they are outside of the scope of this work since we're only focusing on motion data.

A previous study evaluated common objective metrics for gesture generation systems submitted to the GENEA Challenge 2022 [11]. The challenge provided a synchronized dataset of motion, audio, and text with the goal of generating gestures given new input. Submitted results from participating teams were evaluated on large-scale user studies to assess human-likeness and appropriateness to speech [25]. Four objective metrics were applied to every submission and compared to the median ratings of human-likeness from the user study. The first is average acceleration and jerk, the third time derivative of joint positions; the Hellinger distance between speed histograms; the Canonical correlation analysis; and the Fréchet Gesture Distance (FGD), which computes the distance between synthesized and ground truth gesture distributions [24]. However, only the last was reported to have a moderate correlation to human-likeness ratings. Indeed, FGD has been widely employed as a metric to compare motion quality and, thus, as a proxy to human-likeness [16, 22, 28, 29].

Although FGD reportedly correlates with human-likeness ratings, it is unsuitable for tasks such as comparing gesture expressiveness since it focuses on the distance of gesture distributions. On the other hand, aspects of motion dynamics (e.g., speed, acceleration, jerk, among others) have been used to alter perceived expressiveness in virtual characters [14, 15, 18]. It is reasonable to assume that natural-looking synthesized motion has similar values to real human motion. Despite offering some insights, motion dynamics also have limitations. They often analyze individual motion characteristics in isolation, while human perception integrates these dynamics across the entire gesture sequence. Furthermore, they don't provide any intuition on gesture volume, area, or extent. This highlights the need for metrics that capture the holistic nature of gesture communication.

We propose to analyze the area coverage of gesture motion to overcome some limitations of previous metrics. We define the Gesture Area Coverage (GAC) as the temporal grouping of gestures in a motion sequence, i.e., the spatial footprint of gestures throughout the sequence. Our analysis quantifies GAC by mapping a motion

sequence into a raster matrix. We employ two metrics to compare the matrices of different motion sequences: the Dice Score and the Relative Coverage, focusing on the intersection and the difference between the sequences, respectively. In this paper, we show the intuition behind analyzing area coverage and its application for comparing motion in an expressive co-speech gesture dataset, and we analyze its correlation with human-likeness ratings in a large-scale user study. The GAC analysis and code are publicly available at https://github.com/AI-Unicamp/gesture-area-coverage.

## 3 Method

The Gesture Area Coverage (GAC) analysis consists of rasterizing the gesture poses in a motion sequence. Bresenham's line algorithm is used to convert each skeleton bone segment (from a joint to its parent) into a corresponding line segment within a discrete grid, an image. Each non-zero value in the resulting image corresponds to a pixel close to a bone. The GAC of a motion sequence is quantified as the union of the rasterized pose of each frame (as the top images in Figure 2). The total area covered by gestures throughout the motion sequence is calculated as the sum of all elements within the GAC representation (Total GAC).

While evaluating motion sequences, a higher Total GAC for a test sequence than a reference indicates greater overall coverage. However, further insights can be gained by analyzing the distribution of GAC using set theory concepts like True Positive, False Positive, and False Negative. For the present context, True Positive GAC refers to the area covered by both the reference and the test sequence, while True Negative GAC refers to the area not covered by either sequence. False Positive GAC indicates the area extrapolated by the test sequence, i.e., not covered in the reference sequence. False Negative indicates the opposite. The bottom images of Figure 2 provide examples of these sets.

Two metrics encapsulate useful information for evaluating human gestures and comparing motion sequences. The first is the Relative Coverage (RC), defined by the ratio of False Positive (FP) and False Negative (FN) on a logarithmic scale, thus $RC = \log(FP/FN) = \log(FP) - \log(FN)$. Negative values for Relative Coverage suggest a more constrained test sequence, while positive values indicate an extrapolation of the covered area. The second is the Dice Score, a commonly used metric for image segmentation [4], defined as twice the True Positive divided by the sum of the Total GAC of both the reference and test sequences. This metric measures the overlap between the reference and test sequences in terms of covered area. A value closer to one indicates a higher degree of overlap, while values near zero suggest minimal overlap.

We employed these two metrics in two distinct scenarios. The first one consisted of employing GAC to evaluate the ZEGGS dataset. This dataset contains 67 motion sequences categorized into 19 different gesture styles performed by a single professional female actor. We compared every motion sequence of a given style with the neutral.

For the second approach, we applied the metrics in the conditions of the GENEA Challenge 2023 [10]. The challenge provided a large-scale subjective evaluation of gesture generation models from participating teams. 15 conditions were evaluated, the ground truth,

two baselines based on a system from the previous challenge [1, 25], and 12 entries to the 2023 challenge [2, 3, 5, 7–9, 17, 19, 20, 23, 26, 27].

## 4 Results and discussion

Figure 2 shows examples of our approach for comparing two expressive motion sequences with a neutral one. The Dice Score equals two times the medium blue area divided by the sum of the black area of each corresponding sequence. The Relative Coverage equals the light blue divided by the dark blue on a logarithmic scale.

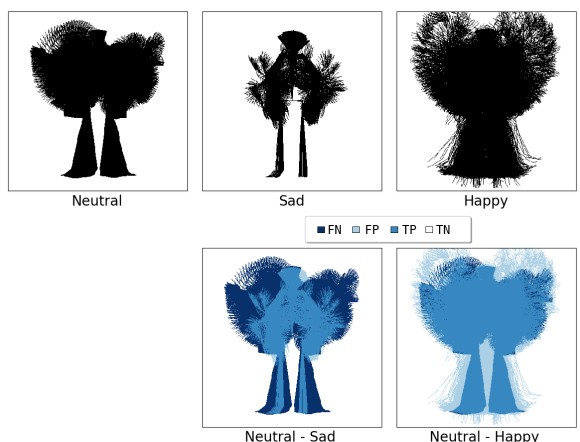

Figure 2: Total GAC of one motion sequence of the Neutral, Sad, and Happy styles from the ZEGGS dataset (top). At the bottom, Sad and Happy are subtracted from Neutral; dark blue represents False Negative (FN), and light blue False Positive (FP). Blue and white represent True Positive (TP) and True Negative (TN), i.e., areas covered and not covered in both sequences, respectively.

Table 1 summarizes the Dice Score, Relative Coverage, and Total GAC for a selection of styles from the ZEGGS dataset (complete data available in the repository). As expected, there is a correlation between Total GAC and Relative Coverage. Lower energy styles, such as Tired, Old, and Sad, cover smaller areas than the neutral baseline. This results in negative RC values, as most of their GAC overlaps with the neutral reference, as can be seen by the True Positive in Figure 2. Conversely, high-energy styles like Happy and Angry exhibit positive values due to a larger deviation from the neutral reference. These findings suggest that the RC is sensitive to variations in motion style, potentially making it useful for applications such as emotion detection, and identity or style classification.

The Dice Score, on the other hand, shows a narrower range across styles: 0.70 to 0.84 with a standard deviation of 0.04, excepting Still. It is reasonable to expect only small variations in Dice Scores in different styles since all sequences comprise real human motion from a single professional performer. While no significant correlation between Dice Score and specific styles was found in this dataset, further investigation with more performers and diverse style sets (e.g., emotions and personality) is needed to assess its role in gesture evaluation definitively.

Table 1: Comparison of GAC metrics between different and neutral styles from the ZEGGS dataset. From left to right: Dice Score, Relative Coverage (RC), and Total GAC ($\times 10^3$ pixels).

| Style | Dice | RC | Total |
|---|---|---|---|
| Neutral | - | - | 52.2 |
| Happy | 0.74 | 4.71 | 85.7 |
| Angry | 0.83 | 1.01 | 58.2 |
| Speech | 0.77 | 0.70 | 60.2 |
| Scared | 0.73 | 0.32 | 47.2 |
| Tired | 0.72 | -0.45 | 46.3 |
| Old | 0.74 | -1.12 | 40.9 |
| Sad | 0.79 | -1.58 | 41.3 |
| Still | 0.32 | -7.06 | 11.6 |

Figure 3 presents the human-likeness median ratings over FGD and the Dice Score on GAC for each condition from the GENEA Challenge 2023. These metrics are also detailed in Table 2, alongside the Relative Coverage. Each condition consists of 70 motion sequences, each lasting approximately one minute, from the challenge's extended test set. GAC-based metrics were extracted using a pair-wise comparison between motion sequences of each condition and the ground truth (condition NA) and taking their average. The FGD was computed by comparing all sequences of each condition against all sequences in the ground truth. Note that the human-likeness evaluation was performed in a subset comprised of 41 motion sequences with an average duration of 9 seconds, while the objective metrics considered the whole set.

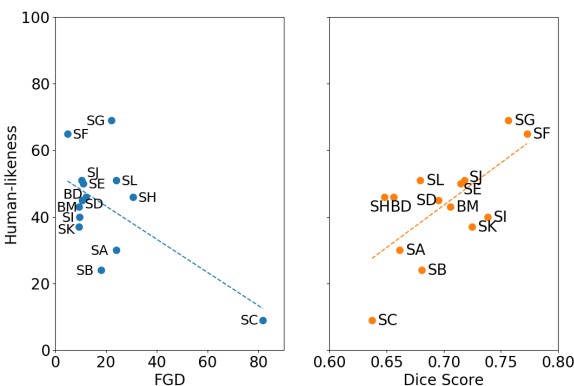

Figure 3: Human-likeness median ratings versus FGD (near 0 is better) and Dice Score on GAC (near 1 is better) of each condition of the GENEA Challenge 2023. SA to SL represent team submissions, and BD and BM are the baselines.

We used Spearman's rank correlation coefficient and its associated statistical test to validate using FGD and Dice Score as alternatives to perceptual evaluations. The results presented in Table 3 show that the Dice Score provided a higher correlation than FGD. However, neither metric achieved a statistically significant correlation ($p < 0.05$).

**Table 2: Comparison of GENEA Challenge 2023 conditions. From left to right: human-likeness median ratings (Hum.), Dice Score, Relative Coverage (RC), and FGD. NA is the ground truth, SA to SL represents team submissions, and BD and BM are the baselines.**

| Cond. | Hum. | Dice | RC | FGD |
|-------|------|------|------|------|
| NA | 71 | - | - | - |
| SG | 69 | 0.76 | -0.21 | 22.1 |
| SF | 65 | 0.77 | -0.47 | 4.9 |
| SJ | 51 | 0.72 | -2.68 | 10.4 |
| SL | 51 | 0.68 | -2.23 | 24.0 |
| SE | 50 | 0.71 | -1.28 | 11.0 |
| SH | 46 | 0.65 | -2.14 | 30.7 |
| BD | 46 | 0.66 | -2.48 | 12.1 |
| SD | 45 | 0.70 | -2.18 | 10.7 |
| BM | 43 | 0.71 | -2.31 | 9.4 |
| SI | 40 | 0.74 | 0.07 | 9.6 |
| SK | 37 | 0.72 | -0.18 | 9.5 |
| SA | 30 | 0.66 | 0.73 | 24.0 |
| SB | 24 | 0.68 | -0.68 | 18.1 |
| SC | 9 | 0.64 | 1.54 | 81.8 |

**Table 3: Spearman's rank correlation of the Dice Score and the FGD to human-likeness median ratings.**

| | Dice | FGD |
|---------|------|------|
| Spearman | 0.47 | -0.17 |
| $p$-value | 0.09 | 0.55 |

From Table 2, negative RC values indicate that gestures are constrained to the ground truth gesture area. However, different than before, since we are dealing with synthetic data, here positive values might indicate erratic and possibly unnatural gestures. This can be seen in Table 2 for conditions SI, SA and SC; their RC are positive while obtaining lower ratings for human-likeness. Most conditions did not achieved positive RC, which is expected since it can be hard for learning-based algorithms to grasp gesture variations fully and most loss functions might restrain results around the average. Although a lower RC indicates that the gesture coverage was smaller than the ground truth, perhaps over-smoothed, it is worth noting that it appears to have little or no correlation with perceived human-likeness, especially given the high rating of conditions SJ and SL.

Although it might be tempting to compare the Dice Scores from the GENEA Challenge 2023 (Table 2) with the ZEGGS dataset (Table 1), we stress that they represent different things. ZEGGS' Dice Scores compare ground truth motion sequences from the same professional performer in different styles and speech, while the other is a pair-wise comparison between ground truth and synthesized gestures from the same speech input.

### 4.1 Limitations

The present analysis focused solely on the front-view plane for gesture coverage. The same core concepts and methods described in this work could be extended to include the top-view plane or

to analyze gesture volume, possibly providing a more comprehensive picture of gesture expressiveness and motion comparisons. Additionally, human gestures often exhibit asymmetries, such as one hand being used more frequently [21]. Future investigations could explore methods to handle these asymmetries. One potential approach is to take the union of the left and right GAC, treating a gesture performed with either hand as valid for both sides.

The current method equally considers every pose of the reference motion sequence. However, gestures tend to be heavily concentrated around a single area [13], as seen in the third pair of images in Figure 1. The Dice Score and the Relative Coverage primarily focus on the overlap in the covered area between the reference and test sequences, potentially overlooking the temporal distribution of poses within the sequence. Thus, the motion sequence might still appear unnatural or lead to a different perceived style or expressiveness. A weighting term might be considered to account for this limitation. Similar considerations are also valid for joint velocities, as shown in the second pair of images in Figure 1.

Motion sequence length can also impact the metrics employed. Short sequences might not provide sufficient area coverage, leading to minimal overlap between sequences. On the other hand, long sequences might produce overly broad areas that fails to provide meaningful information. Further research is needed to estimate this impact and assess an optimal sequence length.

## 5 Conclusion

This work proposes an analysis of Gesture Area Coverage (GAC) and introduces a set of metrics for evaluating human gestures. We explored GAC to quantify gesture expressiveness and its potential relationship with perceived human-likeness. First, we assessed gesture style variations by comparing them to a neutral baseline. Second, we applied the Dice Score to analyze the conditions evaluated in the GENEA Challenge 2023.

The results suggest that the Dice Score on GAC is promising as a complementary metric to existing methods like the Fréchet Gesture Distance (FGD) for gesture evaluation. However, while the Dice Score exhibited a higher correlation with human-likeness ratings than FGD, neither achieved a statistically significant correlation. These findings highlight the potential of GAC but also emphasize the need for further investigation on more diverse and extensive data to assess its efficacy as an evaluation metric.

## Acknowledgments

This study was partially funded by the Coordenação de Aperfeiçoamento de Pessoal de Nivel Superior – Brasil (CAPES) – Finance Code 001. This project was supported by the Ministry of Science, Technology, and Innovations, with resources from Law No. 8.248, of October 23, 1991, under the PPI-SOFTEX program, coordinated by Softex and published as Cognitive Architecture (Phase 3), DOU 01245.003479/2024-10; by the São Paulo Research Foundation (FAPESP) under grant #2020/09838-0 (BI0S - Brazilian Institute of Data Science); by the Eldorado Research Institute and by the Artificial Intelligence Lab., Recod.ai.

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
