# OpenReview forum: "Gesture Area Coverage to Assess Gesture Expressiveness and Human-Likeness"
_ACM.org/ICMI/2024/Workshop/GENEA — GENEA Workshop 2024_

### Official Review · Reviewer_sauE · 2024-07-26
**A well-motivated and intuitive new evaluation method**

**Rating:** 7
**Confidence:** 4

**Review:**

### Overview

This paper analyses Gesture Area Coverage (GAC) and introduces a set of metrics to evaluate human gesture motion. To my understanding, these metrics are meant to assess human likeness and be independent of appropriateness. It finds these metrics are potentially promising gesture evaluation methods, with Dice Score exhibiting a higher correlation to subjective human-likeness ratings than FGD, albeit not a statistically significant correlation.

### Strengths

This is a well-motivated and novel approach to gesture evaluation.

The proposed metric shows a level of correlation to subjective human evaluation.

The approach is intuitive and easy to understand.

The paper is well-written and concise.

### Questions/Weaknesses

1) The caption for Figure 2 is missing a true positive description.

2) Authors mention that “poses tend to be heavily concentrated around a single area”. [1] defines a peripheral gesture space as containing any gestures performed above the neck and outside of the neck and shoulders, and [2] statistically derives an extreme gesture space as containing a pose more than a standard deviation from the mean pose. Some further analysis may be needed to determine how the measure is affected based on the percentage of time poses are in a specific gesture space, which may be influenced by gesture style or speaker identity.
[1] McNeill, David. "Hand and mind1." Advances in Visual Semiotics 351 (1992).
[2] Windle, Jonathan, et al. "Arm motion symmetry in conversation." Speech Communication 144 (2022): 75-88.

3) The NA, ground truth sequence was compared against the GENEA results. As this is proposed as a human-likeness evaluation method, could using the complete set of poses available for the speaker (e.g. from the training and/or test data) be a better condition to compare against if only interested in human-likeness? Using the same NA sequence may impart some appropriateness element to the metric, albeit not temporal appropriateness.

### Rating

This is a well-written and potentially valuable paper for the gesture evaluation community. More research questions and experiments are needed to determine its efficacy as an evaluation metric. Still, the idea is novel and a step in the right direction.

**Nominate For A Reproducibility Award:**

If code is released as promised, this could be easily reproduced

---

### Official Review · Reviewer_QUfX · 2024-07-26
**Interesting contribution on a potential gesture metric**

**Rating:** 7
**Confidence:** 5

**Review:**

Overall, I think this paper offers an interesting contribution.  Metrics for gesture synthesis would be of great value, so research in this area is useful.  Previous work has used the spatial distribution of gesture as an ad hoc metric (e.g. Fig. 1 in [1]), but this work advances the approach considerably by defining numeric metrics and evaluating both how these metrics reflect style and how they measure synthesis quality.  The correlation with human ratings is still not high enough that the approach could be a substitute, but it does appear to have value and it's useful to see that it outperforms FGD here.  I can see this leading to the development of improved spatial metrics.

It seems that the code will be released, which will increase the value of the work.  This should be clarified.

A limitation of the approach is that a pixel is coded as occupied or not, whereas in reality some pixels may be gestured over a lot and some only once.  The paper discusses this and it seems important to take this frequency into account. Related to this, it seems that the length of the sequence could impact the matric given the current calculation.  Sequences that are too short may not overlap well, even if they are similar.  For longer sequences, there is an increasing danger of an outlier gesture that would cover areas that are rarely used, but would then be treated equally as the commonly used areas with the current construction of the metric.  It would be worth discussing the impact of sequence length.

A paragraph should not start with "Moreover" as the word creates a connection with the previous statement, emphasizing the point made. At the start of a paragraph, there is no previous statement to connect to.

Is there too little data for using FGD?  What size of distribution is required?

1. Ferstl, Y., Neff, M., & McDonnell, R. (2020). Adversarial gesture generation with realistic gesture phasing. Computers & Graphics, 89, 117-130.

**Nominate For A Reproducibility Award:**

I'm not clear if the code will be released.

---

### Official Review · Reviewer_bcYc · 2024-07-26
**The paper "Assessing Gesture Coverage Area: A Practical Approach" explores the assessment of gesture coverage area. The methodology analysis techniques, supported by practical examples and case studies.**

**Rating:** 7
**Confidence:** 4

**Review:**

The paper is well written and it addresses a critical need in the design and implementation of human gesture generation systems. By focusing on gesture coverage, the paper contributes to improving gesture understanding. The methodology is well-structured and detailed, comprising data, analysis techniques, and evaluation metrics. Practical examples and case studies strengthen the paper. The results are clearly presented and aligned with the study's objectives.
To enhance the paper, the authors could consider Include more diverse and extensive data to strengthen the generalizability of the findings. Conduct more extensive user studies to gather feedback and refine the practical approach.

**Nominate For A Reproducibility Award:**

No

---

### Decision · Program_Chairs · 2024-07-30

**Decision:**

Accept

**Comment:**

This concise and well-written paper describes several new automated metrics of gesticulation, contributing towards a more nuanced suite of easy-to-compute objective measures that can be used to obtain a quantitative impression of gesture generation and how it compares to recordings of human gesture motion. The ideas are novel and they consider an important problem. None of the metrics require that any machine learning be performed, which helps make them fast and easy to apply.

All three reviewers considered this a good contribution and recommend accepting this contribution. This meta-reviewer has also read the paper and agrees with the reviewers’ consensus.

The reviewers make several good points, and the authors are encouraged to read and consider the reviews in their entirety. Here are some comments on points that the authors may wish to consider when preparing their final version of the paper:
* Reviewer bcYc mentions the value of additional benchmarks. Why not evaluate the metric also on the GENEA Challenge 2022 data? This seems simple to do and authors may wish to consider it. The article does not seem to discuss why GENEA Challenge 2022 data was not leveraged in the existing evaluation.
* Reviewer QUfX discusses the impact of segment length on area coverage metrics. (As each segment considered grows longer, the space covered by the gestures can only increase.) This seems salient, and the authors may consider adding a note and an explicit discussion regarding this aspect.
* Reviewer sauE notes that, if comparisons are performed between matching segments, then the metrics may be expected to carry an element of appropriateness evaluation as well, and not merely human-likeness (which is independent of permuting the sequences). This meta-reviewer got the same impression from reading the article. It is recommended to clarify this in the text. If the authors wish to go further, they could compare their current numerical results to results obtained by randomly mismatching NA sequences against other sequences from the condition being assessed.

One additional thought: Would it not be better to propose to consider the FP/FN numbers on a logarithmic scale (instead of on a linear one, as now), i.e., compute log(FP/FN) = log(FP) - log(FN)? Using logarithmic scales is commonplace for values that involve multiplication or division, helps compress the dynamic range, and would make it easier to eyeball the numbers. Right now, adding 0.1 to the value for Still in Table 1 would make a major difference, but adding the same number to Happy would essentially make no difference at all. In other words, the ticks on the FP/FN scale are not uniform, and instead proposing to consider the logarithm of the ratio would go a long way towards addressing that.

Additional minor corrections:
* Line 253: "Neutral is subtracted by Sad and Happy" might be better written as "Sad and Happy are subtracted from Neutral"
* Line 378: "ZEEGS’" -> "ZEGGS’"